

# On the self-similarity of wind turbine wakes in complex terrain using large-eddy simulation

Arslan Salim Dar[1,2], Jacob Berg[1], Niels Troldborg[1], and Edward G. Patton[3]

[1]DTU Wind Energy, "Risø campus", Frederiksborgvej 399, 4000 Roskilde, Denmark
[2]Wind Engineering and Renewable Energy Laboratory (WIRE), École Polytechnique Fédérale de Lausanne (EPFL), 1015 Lausanne, Switzerland
[3]National Center for Atmospheric Research, Boulder, CO, US

**Correspondence:** Jacob Berg (jbej@dtu.dk)

**Abstract.** We perform large-eddy simulation of flow in complex terrain under neutral atmospheric stratification. We study the self-similar behavior of a turbine wake as a function of varying terrain complexity and perform comparison with a flat terrain. By plotting normalized velocity deficit profiles in different complex terrain cases, we verify that self-similarity is preserved as we move downstream from the turbine. We find that this preservation is valid for a shorter distance downstream compared to what is observed in flat terrain. A larger spread of the profiles toward the tails due to varying levels of shear is also observed.

## 1 Introduction

Rotor wakes have a consequential impact on the efficiency of a wind farm, as the turbines standing in wake generally face lower wind speeds, along with, enhanced turbulence levels (Barthelmie et al., 2007). The particular dynamics of wakes are strongly influenced by the terrain characteristics, such as, surface vegetation and sloping terrain. Although, a lot of emphasis has been on understanding wakes in flat terrain over the past decade (Medici et al., 2006; Jimenez et al., 2007; Chamorro and Porté-Agel, 2009; Iungo et al., 2013; Calaf et al., 2010; Porté-Agel et al., 2011; Abkar et al., 2016; Allaerts and Meyers, 2015; Iungo, 2016), complex terrain now finally gets the attention it deserves. This is partly due to a prospective shift in development of wind farms from flat to complex terrains caused by saturation of ideal flat terrains and partly due to the recent developments observationally and numerically. Understanding wakes from turbines in complex terrain, therefore, becomes important for understanding the interaction between terrain and wakes, as well as, for better resource assessment and wind farm siting. The change in topography gives rise to flow phenomenon such as speed up across the hills, flow separation behind hills with high slopes and generation of localized turbulent structures. This makes flow prediction in complex terrain challenging and turbine behavior in such locations is far from understood.

Recent works on wake interaction in complex terrain either deal with the topic in ideal complex terrains such as Gaussian or sinusoidal hills, or present site specific studies. Tian et al. (Tian et al., 2013) performed an experimental study of five turbines





located on a Gaussian two-dimensional hill to understand the interaction of wind farm with the terrain. Their experimental setup was numerically regenerated in an LES domain by Shamsoddin and Porté-Agel in an attempt to validate their LES model (Shamsoddin and Porté-Agel, 2017). Hyvarinen and Segalini also studied wakes over periodic sinusoidal hills (Hyvarinen and Segalini, 2017). Using experimental and numerical tools, they achieved good agreement between the two, however,

implementing the Jensen wake model (Katic et al., 1986) did not yield good results.

Coming to the site specific studies, Castellani et al. studied the impact of wakes and terrain on the performance of a wind farm located in Southern Italy (Castellani et al., 2017). Lutz et al. used Detached Eddy Simulation to study the wake from a single turbine in a complex site in Germany (Lutz et al., 2017). The impact of topography on wake development was highlighted by comparison with a flat terrain case. Recently, Berg et al. performed a Large-Eddy Simulation study of wake from a wind

turbine located at the site of Perdigão, Portugal (Berg et al., 2017) and highlighted some differences in wake orientation from Shamsoddin and Porté-Agel's work (Shamsoddin and Porté-Agel, 2017). From analysing wind scanner data from Perdigão, Menke et al. (Menke et al., 2018) show a strong influence of atmospheric stability on wake propagation and orientation. This further supports the hypothesis that it is difficult to generalize wind turbine wake results for complex sites.

Wakes behind turbines are known to show self-similar behavior in a flat terrain under different atmospheric conditions. Xie

and Archer showed mean velocity deficit behind turbines to be self-similar in neutral conditions (Xie and Archer, 2014). Abkar and Porté-Agel found self-similarity under various stability classes (Abkar and Porté-Agel, 2015). This was further explored by Xie and Archer, who then checked for self-similarity under different stability conditions in the presence of Coriolis force (Xie and Archer, 2017). This self-similar behavior of mean velocity deficit in the far wake has been a fundamental assumption for many analytical wake models (Katic et al., 1986; Xie and Archer, 2014; Abkar and Porté-Agel, 2015; Bastankhah and F.

Porté-Agel, 2014).

In the current work, we look for self-similarity of wind turbine wakes in complex terrain under neutral atmospheric conditions and without the effect of Coriolis force. For this purpose, we extend the work by Berg et al. (Berg et al., 2017) and verify self-similarity of wakes under different terrain characteristics and turbine locations. If successful, this can potentially provide basis for the development of analytical models for wakes in complex terrain (Luzzatto-Fegiz, 2018). This self-similarity of

rotor wakes in complex terrain has not yet been verified in the existing literature to the best of the authors' knowledge.

The rest of the paper is structured as follows: section II details the LES code used for the study, terrain characteristics along with case configuration and data description is provided in section III. A nomenclature for wake statistics necessary for self-similarity check is presented in section IV, whereas, results from the study are shown in section V. The paper is finally concluded in section VI.





## 2 LES Framework

The large-eddy simulation code used in the current study is formerly formulated in (Sullivan et al., 2014) with examples of usage in complex terrain in (Sullivan et al., 2010). The governing equations are the spatially filtered incompressible Navier-Stokes and continuity equations, which for a neutrally stratified atmospheric flow read:

$$\frac{\partial \tilde{u}_i}{\partial x_i} = 0 \tag{1}$$

$$\frac{\partial \tilde{u}_i}{\partial t} + \frac{\partial \tilde{u}_i \tilde{u}_j}{\partial x_j} = -[\frac{\partial \tilde{p}^*}{\partial x_i} + \frac{f_i}{\rho} + \frac{\partial \tau_{ij}}{\partial x_j}] + F_p \delta_{i1} \tag{2}$$

Where $\tilde{u}_i$ is the resolved velocity in $x$, $y$ and $z$ directions corresponding to i = {1, 2, 3} respectively. The pressure variable $\tilde{p}^*$ is solved using the elliptic Poisson equation in an iterative manner. $f_i$ is the body force which models the interaction of
the turbine with the flow and $\rho$ is the fluid density, which is kept constant throughout the study. The kinematic sub-grid scale stresses are represented by $\tau_{ij} = \widetilde{u_i u_j} - \tilde{u}_i \tilde{u}_j$. The external forcing driving the flow is represented by $F_p$, which in current case is the constant pressure gradient in the streamwise direction (see later). An important thing to note is that the Coriolis and buoyancy forces are neglected in the current study.

A terrain following coordinate transformation is used to represent the complex geometry. The transformation that maps
physical coordinates $x_i = (x, y, z)$ onto computational coordinates $\xi_i = (\xi, \gamma, \zeta)$, $x_i \Rightarrow \xi_i$ is given by the rule:

$$x = x(\xi) = \xi \tag{3}$$

$$y = y(\gamma) = \gamma \tag{4}$$

$$z = z(\xi, \gamma, \zeta), \tag{5}$$

with the corresponding Jacobian is given by:

$$J = det \begin{bmatrix} \xi_x & 0 & 0 \\ 0 & \gamma_y & 0 \\ \zeta_x & \zeta_y & \zeta_z \end{bmatrix} = \xi_x \gamma_y \zeta_z = \zeta_z \tag{6}$$

where, the subscript denotes partial derivative. The governing equations, (1) and (2), can now be transformed using the chain rule and the identity (consult (Sullivan et al., 2014) for details):

$$\frac{\partial}{\partial \xi_j} \left( \frac{1}{J} \frac{\partial \xi_j}{\partial x_i} \right) = 0, \tag{7}$$

in order to write a set of equations in a strong flux-conservation form using the volume flux variables,

$$U_i = \frac{\tilde{u}_j}{J} \frac{\partial \xi_i}{\partial x_j}, \tag{8}$$





so that $U_i = (U, V, W)$ are normal to surfaces of constant $x_i$. In the numerical mesh $W$ is located on cell faces while $U$ and $V$ are located in cell centers. This ensures a tight coupling to pressure defined at the cell centers. Solving the Poisson equation involves an iteration procedure which incurs additional expense compared to conventional computations in flat terrain (again we recommend the reader to consult (Sullivan et al., 2014) for details).

The code is pseudo-spectral code, with wave number representation in the horizontal directions, $(\xi, \gamma)$ and second order finite-difference in the vertical direction, $\zeta$.

The physical ($z$) and computational ($\zeta$) vertical coordinates are related to each other using a simple transformation rule, with $z$ exponentially stretched from the surface:

$$\zeta = Z_L \frac{z - h_L}{Z_L - h_L} \qquad (9)$$

where, $h_L$ is the local terrain height and $Z_L$ is the total height of the computational space.

The Deardorff SGS model as adopted in (Moeng, 1084; Sullivan et al., 1994) is implemented to parameterize the sub-grid scale stresses.

## 2.1   Turbine Parameterization

An actuator disk model without any rotational effects is implemented to represent the turbines. The unavailability of a well

defined free stream velocity $U_\infty$ in complex terrain is compensated by employing the classical expression given by (Hansen, 2015):

$$U_\infty = \frac{\langle u_d \rangle}{1 - a} \qquad (10)$$

where, $\langle . \rangle$ denotes ensemble averaging (note that in current work, ensemble averaging is same as time averaging), $\langle u_d \rangle$ is the velocity at the rotor disk and $a$ is the axial induction factor. The model simulates turbines using disks with a thrust force given

by:

$$F_t = -\frac{1}{2} \rho C_T' \langle u_d \rangle^2 \frac{\pi}{4} D^2 \qquad (11)$$

where, $D$ is the rotor diameter and $C_T'$ is related to the thrust coefficient $C_T$ using one-dimensional momentum theory:

$$C_T' = \frac{C_T}{(1 - a)^2} \qquad (12)$$

For the current work, the chosen values for $a$ & $C_T$ are 1/4 & 3/4 respectively, which yield a $C_T'$ value of 4/3. The use of

$\langle u_d \rangle$ was (to the authors knowledge) introduced into atmospheric LES by (?). In the numerical code, the thrust force is then distributed proportionally to the fractional area of each cell covered by the rotor. The force per unit volume in the longitudinal direction is then given by:

$$f_x(i, j, k) = -\frac{1}{2} \rho C_T' \langle u_d \rangle^2 \frac{\Lambda(j, k)}{\Delta x} \qquad (13)$$

where, $\Lambda$ is the fractional area and $\Delta x$ is length of a given cell.



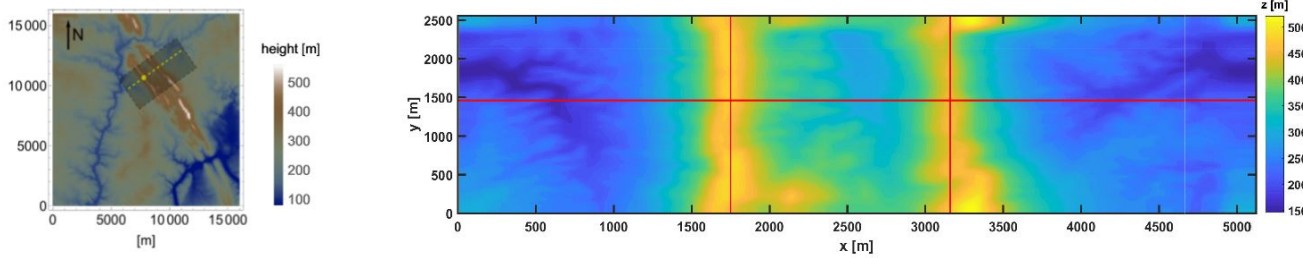

**Figure 1.** Terrain elevation variation at Perdigão. (Left) Shaded rectangle represents the simulated area, with yellow line passing through the main transect of turbine location (source: (Berg et al., 2017)), (right) shaded region from left figure with intersection of red lines representing the turbine locations.

## 3 Case Configurations

Simulations are performed on a domain spanning over $5120 \times 2560 \times 3000 \ m^3$, which represents a double ridge configured site named Perdigão in Portugal (Vasiljević et al., 2017). The site has been a focus of intensive field measurement campaign under the Perdigão experiment 2017 (Wildmann et al., 2018; Menke et al., 2018) and the New European Wind Atlas (NEWA) project
(Mann et al., 2017). Figure 1 shows the overview of the site, where, the wind direction is along the main transect (235 degrees relative to North). With its double ridge configuration and challenging slopes, the terrain is ideal for a comprehensive study of interaction between wind turbine wake and topography. The atmosphere is assumed to be neutrally stratified and boundary conditions used are described below. A roughness length of $z_0 = 0.5$ m is chosen keeping the rugged, forested terrain in mind, giving $z_0/H = 1.67 \times 10^{-4}$. The simulations are initiated from a random incompressible velocity field and run for a
time of $T_S = 100 T_E$ until stationarity was achieved. Here, $T_E = H/u_*$ ($u_*$ is defined as friction velocity) is the time scale corresponding to the size of the largest eddy that can possibly fit in the domain and is based on a simple momentum transfer argument borrowed from the flat terrain. Figure 2 shows an instantaneous snapshot of streamwise velocity along the main transect passing through the turbine. The turbine is assumed to have a rotor diameter and hub height of 80 meters each. This is done to somewhat match the dimensions of an Enercon 2 MW turbine actually deployed at the upwind ridge at Perdigão.

The iteration procedure performed when solving the Poisson equation is expensive due to the strong coupling between pressure and vertical wind component and the fact that in contrast to flat terrain, pressure gradients are quite high in complex terrain and thus require more time to fully converge. Slight smoothing of the terrain can help in reducing the computation time by reducing the number of pressure iterations required to attain convergence. This is done by applying an exponential filter in the
wave number ($x$ & $y$) space. This filtering process is given as follows:

$$h^{model}(x,y) = \int\int h^{true}(k_x, k_y)\tilde{F}(k_x, k_y)e^{i(k_x x + k_y y)}dk_x dk_y \tag{14}$$



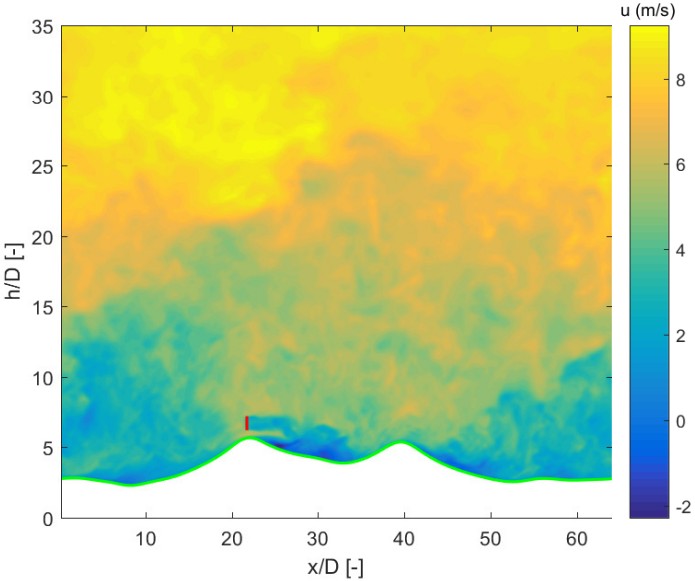

**Figure 2.** Instantaneous velocity in the streamwise direction along the main transect with turbine (in red).

where, $\tilde{F}(k_x, k_y)$ is the exponential filter given by:

$$\tilde{F}(k_x, k_y) = e^{-\alpha[(k_x \Delta x)^2 + (k_y \Delta y)^2]} \tag{15}$$

here, $\alpha$ is the control parameter which determines the level of smoothing. Two different values of $\alpha$ used in the current study are 4 & 0.5, which result in maximum terrain slope of 0.57 & 0.77 across the main transect respectively. The steep terrain nearly matches the original terrain. The $\alpha$ value of 0.5 instead of 0 is chosen for this terrain to avoid Gibbs phenomenon. Figure 3 shows a comparison of two terrains at a fixed lateral position passing through the turbine location. From here onward, terrain with maximum slope of 0.57 will be referred to as 'smooth', whereas, one with maximum slope of 0.77 will be referred

to as 'steep'.

Table 1 gives an overview of various cases analyzed in the current study. It is to be noted that the number of ensembles used is fairly high to guarantee stationarity and an apparent difference in the number of ensembles for different cases does not affect the results. The aspect ratio of 1:4 in the case of 20 m resolution lies towards the limit of the allowed aspect ratio for the

15 chosen sub-grid scale model (rule-of-thumb).



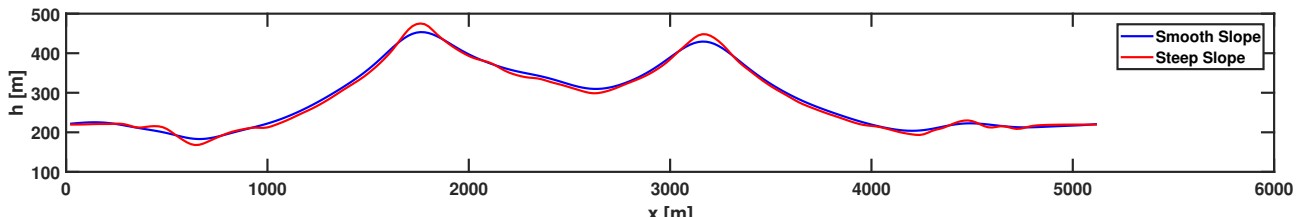

**Figure 3.** Comparison of terrains with two different slopes along the main transect

| Case No. | Grid | $\Delta x$ & $\Delta y$ (m) | $\Delta z$ (m) (at lower boundary) | No. of Turbines | Terrain (Slope) | No. of 30 min avg. ensembles (w/ Turbine) & (w/o Turbine) |
|---|---|---|---|---|---|---|
| 1 | $256 \times 128 \times 128$ | 20 | $\approx 5$ | 1 | Smooth (0.57) | (44) & (43) |
| 2 | $256 \times 128 \times 128$ | 20 | $\approx 5$ | 1 | Steep (0.77) | (45) & (50) |
| 3 | $256 \times 128 \times 128$ | 20 | $\approx 5$ | 2 | Steep (0.77) | (46) & (50) |
| 4 | $512 \times 256 \times 128$ | 10 | $\approx 5$ | 2 | Steep (0.77) | (35) & (35) |
| 5 | $256 \times 128 \times 128$ | 20 | $\approx 5$ | 1 | Flat (0) | (68) & (68) |

**Table 1.** Description of cases analyzed in the study

## 3.1 Boundary Conditions

The boundaries in the horizontal directions are fully periodic, where, a constant pressure gradient given by $dP/dx = -u_*^{+2}/H$ is applied in the x-direction to drive the flow. Here, $u_*^+$ is the friction velocity chosen to be 0.6 m/s and $H$ is the height of domain equal to 3000 m. It is worth mentioning here that due to complexity of terrain, the value used for friction velocity is

5    not the effective friction velocity in the domain (will be discussed later). The periodicity in the horizontal directions means that the terrain is repeating itself, i.e. the inflow is affected by the terrain in the domain itself. We have made tests with extended buffer regions (not shown) which indicated that the turbulence level in the incoming flow was not affected by the terrain itself, and we therefore do not consider the periodicity to be a shortcoming of the conclusions later to be made.

10   The lower surface of domain is modelled assuming log-law behavior in the first cell as a point-wise implementation (Bou-Zeid et al., 2004). A no stress condition is applied at the upper boundary of the domain. Under this condition, the gradients of horizontal velocity components are set to zero, whereas, vertical velocity component itself is set to zero. Moreover, the sub-grid scale energy ($e$) is also set to zero at upper boundary.



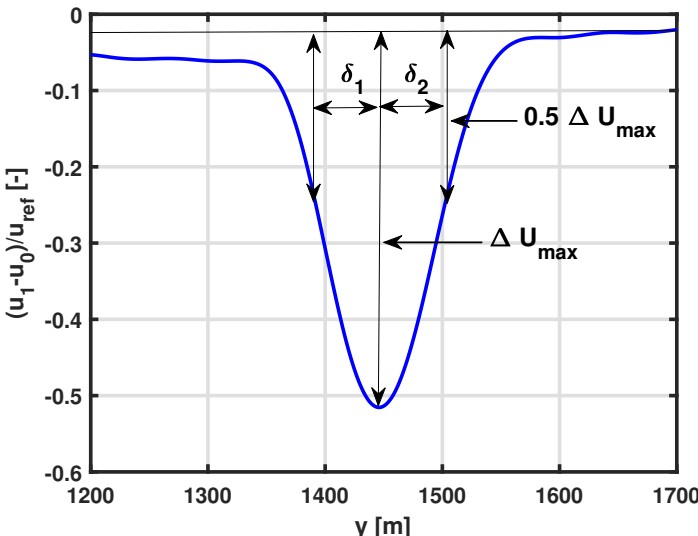

**Figure 4.** Nomenclature for wake half-width.

### 3.2 Data Description

We base our analysis on averaged LES data, a choice associated with the need of stationary flow characteristics required for evaluation of self-similar wake statistics. We save 30 minutes averaged LES fields and by averaging a number of these fields together, a total time necessary to guarantee stationarity is obtained. The rule for this averaging is defined as follows:

$$\langle u \rangle = \frac{1}{N} \sum_{n=1}^{N} u_{(30,n)} \tag{16}$$

where, $\langle u \rangle$ is the three dimensional ensemble averaged velocity field, $u_{(30,n)}$ is the $n^{th}$ 30 minute averaged field and $N$ is the total number of 30 minute averages used, as given in table 1 for each case. To simplify the notations, mean streamwise velocity computed using above rule will be denoted by $u = \langle u \rangle$.

### 4 Wake Flow Nomenclature

The quantities required to characterize the wake flow for self-similarity check are defined in the current section. We first define the normalized mean velocity deficit as follows:

$$\Delta U = \frac{u_1 - u_0}{u_{ref}} \tag{17}$$



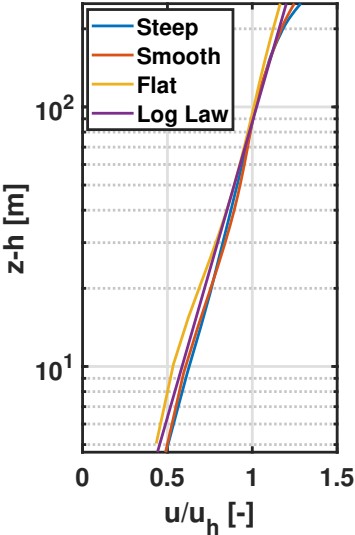

**Figure 5.** Comparison of normalized mean streamwise velocity at $x = 1$.

where, $u_1$ is the averaged streamwise velocity in the simulation including the turbine, $u_0$ is the same in the absence of the turbine. The mean difference is then normalized with a reference velocity, which corresponds to the velocity at the hub height of the first turbine in the absence of the turbine.

Wake half-width is defined as the distance from wake-center to the point where the velocity deficit is reduced to 50 % of the
maximum velocity deficit. As wind turbine wakes can be asymmetric, the wake half-width is computed for two sides of the wake individually. The nomenclature used when describing the wake characteristics is defined in figure 4. The wake centerline is traced by identifying the point of maximum velocity deficit at each downstream location.

## 5 Results

### 5.1 Inflow Velocity

To characterize the incoming flow velocity, normalized mean velocity profiles at the inflow for different cases along with a logarithmic profile are shown in figure 5. Here, $u_h$ refers to the velocity at the hub height of the first grid point in the streamwise direction. It can be observed that the simulated velocity profiles show some deviation from the logarithmic profile, especially in the case of complex terrain. These deviations are somewhat expected, as the logarithmic profile is not completely valid in heterogeneous terrain. Moreover, the deviations observed are due to differences in the slopes of various profiles, whereas, the
quantitative values lie very close to the logarithmic profile. The periodic boundary conditions also play a role in determining the inflow velocity and therefore could be responsible for some of the deviations from the logarithmic profile.





## 5.2   Impact of Grid Resolution

The steep slope case is simulated at two different horizontal grid resolutions of 20 m and 10 m respectively. Figure 6 shows vertical profiles of streamwise velocity across the turbine located on top of first ridge for the two grid resolutions. Considering the relative positioning of rotor in the two cases, the velocity profiles are observed to be in good agreement with each other.

However, it is important to note that the agreement between velocity profiles for the two cases is sensitive to the chosen reference velocity. Figure 7 shows a comparison of streamwise velocity along the main transect at the hub height, as well as, local slopes for the two grid resolutions. Whereas, the qualitative trend of the velocity profiles match well for the two resolutions, the quantitative values differ by up to 15% between the two cases. This apparent discrepancy can be somewhat justified by the change in terrain characteristics caused by the change in resolution. As the resolution gets finer, the terrain

becomes more detailed and the slopes are slightly changed. This can be verified by comparing the rotor positions in the up and down-stream of the ridge.

Recall that in section 3.1 the flow was defined to be pressure gradient driven with a friction velocity of 0.6 m/s. This balance between pressure gradient and friction velocity would hold true in flat terrain, whereas, in complex terrain there is an additional contribution from the form drag generated by the two ridges. The immediate impact of this additional contribution is a reduced

friction velocity in the terrain. As the terrain gets more detailed in the finer grid, the friction velocity is further decreased by virtue of an increased contribution from form drag. We believe that this difference in friction velocity in the two grid resolutions can explain the difference in the velocity profiles observed here.

A comparison of the LES flow characteristics in different terrains and resolution with field measurements, although without any turbine in the terrain, is recently presented by Berg et al. (Berg et al., 2018). Good agreement between the measurements

and simulations is obtained, keeping in mind the simple flow conditions defined in the simulations.

## 5.3   Wake Characteristics

A fundamental question can be formulated around the interaction of wake from the turbine with the local terrain. Understanding this kind of interaction is important in order to know how the wake behaves within a complex terrain and how the terrain generated phenomenon impact the behavior of the rotor wake. Figure 8 (a) shows the wakes from turbines in two different

terrains and two different locations (top of the respective ridges). An important impact of local topography on wakes is the change in the orientation of wakes behind the three turbines. The strong recirculation regions developed behind the ridges due to flow separation generate their own shear layers which do not allow the turbine wakes to mix with them. It is for this reason that we observe an upward orientation for the turbines located on top of the first ridge in the two cases. For the smooth terrain, the recirculation starts on top of the ridge and the height of this region extends higher than the ridge top. This causes an initially

horizontal wake to shift in an upward direction. For the steep case, however, the high slopes result in an upward inclination of the wake from the rotor position. For the turbine located on the second ridge, the wake is oriented in a slightly downward, somewhat, terrain-following orientation. This is due to the fact that the recirculation region behind the second ridge is smaller than the ridge top and thus allows the wake to move in a downward direction.



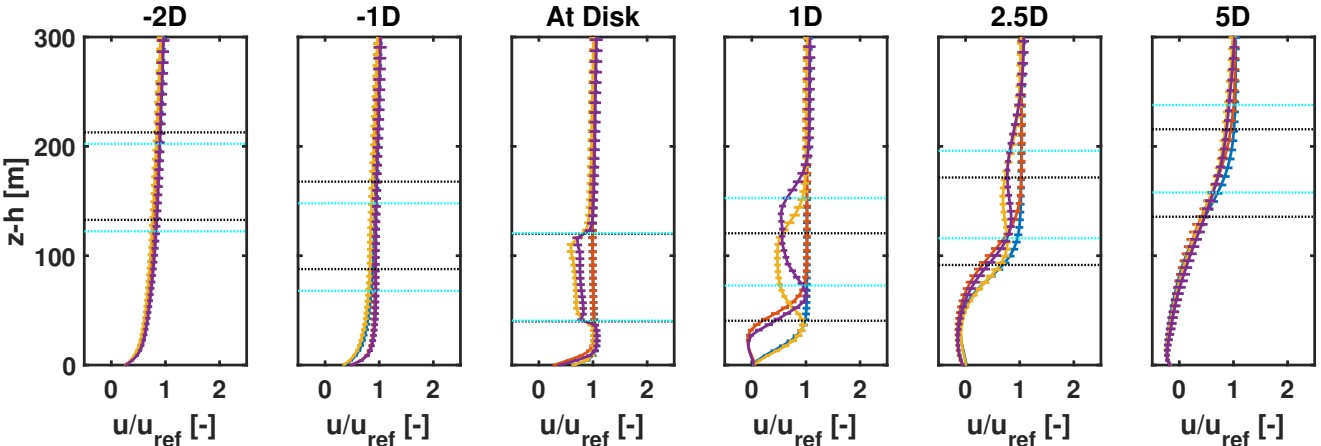

**Figure 6.** Vertical profiles of normalized mean streamwise velocity around first ridge for two different grid resolutions. Blue and yellow vertical lines show velocity profiles without and with turbine for 20 m resolution, whereas, red and purple vertical lines show profiles without and with turbine for 10 m resolution. Black horizontal lines trace rotor top and bottom for the coarse grid resolution, whereas, cyan horizontal lines trace the same for finer grid resolution. The error bars correspond to $\sigma/\sqrt{N}$, where, $\sigma$ is the standard deviation and N is the number of 30 minute ensembles used.

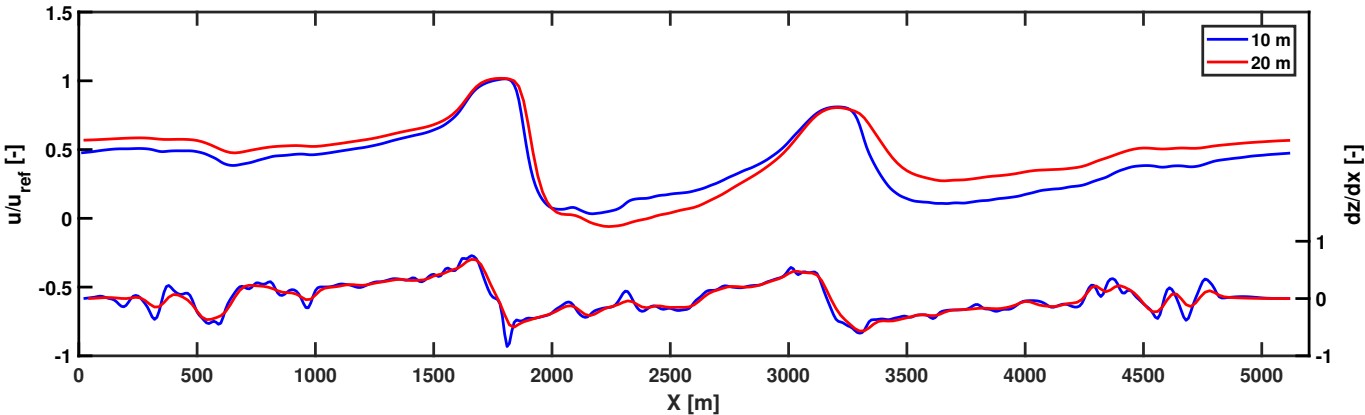

**Figure 7.** Comparison of streamwise velocity at the hub height without turbine and local slopes along the main transect for the two grid resolutions.

Figure 8 (b) shows wake development in the rotor plane. The sideways movement of wake can be observed in the three cases, as the wake moves downstream. Although, the most probable reason for this sideways movement is the terrain complexity and variation in pressure distributions across the terrain, any conclusive statement cannot be made due to the limitations of grid resolution employed in the current study.



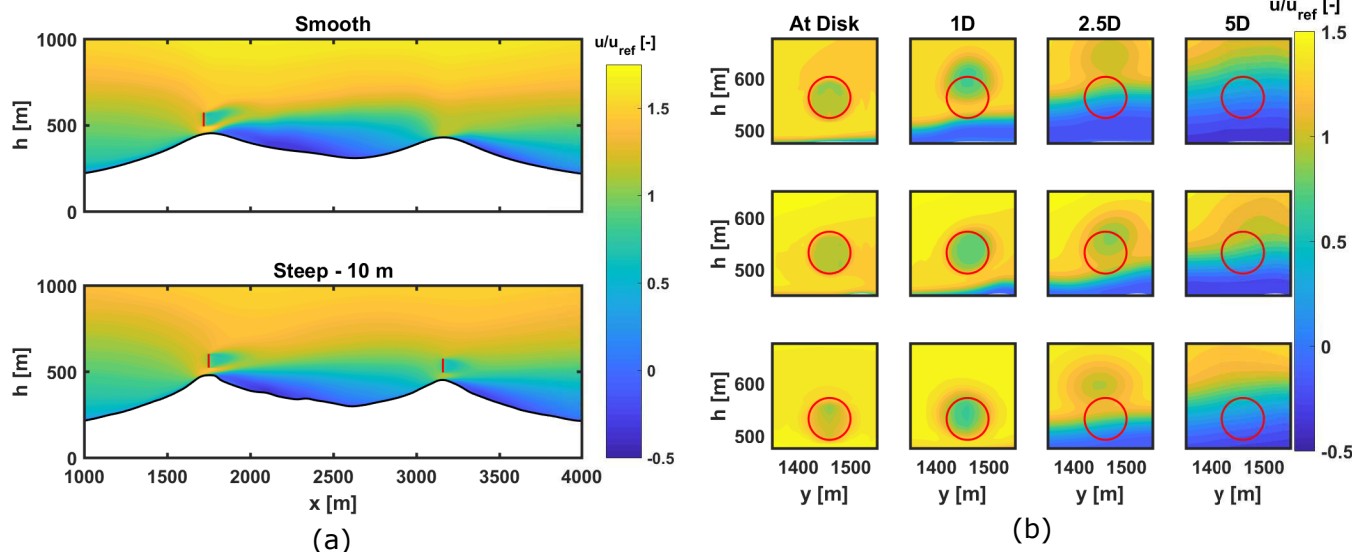

**Figure 8.** (a) Streamwise velocity profile for smooth and steep terrain, (b) Wake development in the rotor plane at different downstream locations. First row: 1st turbine, steep terrain - 10 m resolution; second row: 2nd turbine, steep terrain - 10 m resolution; third row: 1st turbine, smooth terrain. Rotor position is highlighted by the red circle.

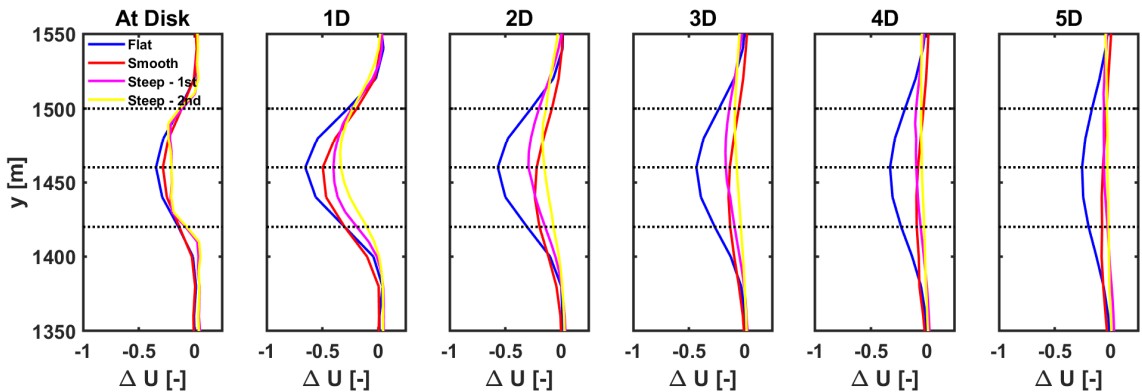

**Figure 9.** Lateral profiles of streamwise velocity deficit for different cases.

An important feature of the wake is the relatively faster recovery as compared to the flat terrain. High levels of turbulent mixing in the atmosphere due to complexity of the terrain provide a catalyst effect and thus promote quick recovery of wake. It is, thereby, observed that the wake is almost recovered at a distance of around 5 rotor diameters in the downstream. This faster wake recovery in the complex terrain has been previously reported by various other studies (Politis et al., 2012; Astolfi et al., 2018; Tabib et al., 2016).



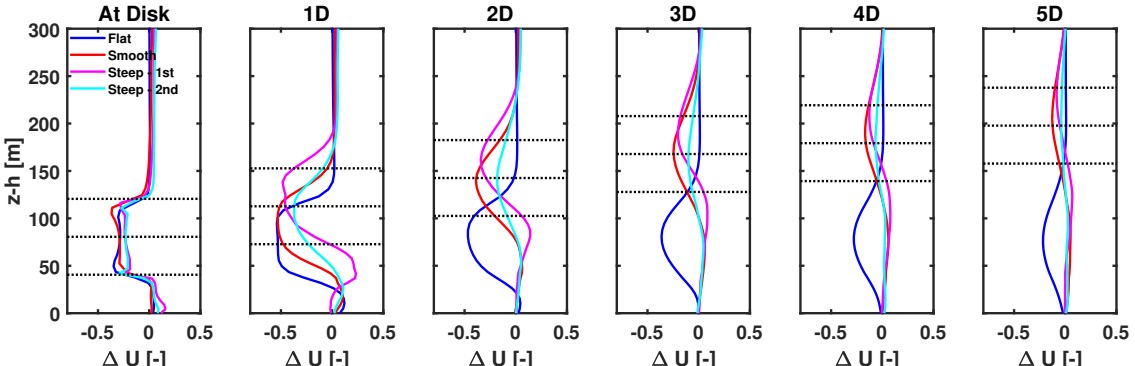

**Figure 10.** Vertical profiles of streamwise velocity deficit for different cases.

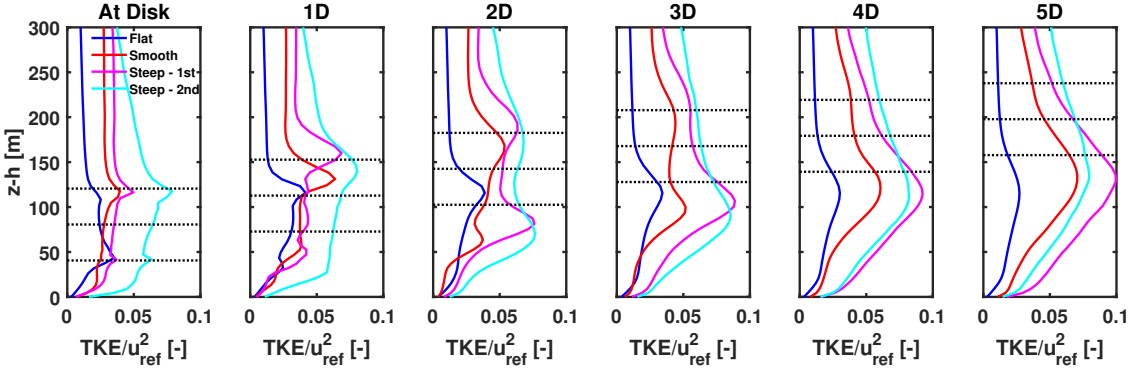

**Figure 11.** Vertical profiles of normalized turbulent kinetic energy for different cases.

### 5.3.1 Velocity Deficit Profiles

Figures 9 & 10 show normalized mean velocity deficit profiles for different cases (flat, smooth, steep - 10 m & 1st turbine, steep - 10 m & 2nd turbine) at different downstream locations. Comparing the lateral profiles, it is observed that the finer grid resolution in the lateral direction captures the wake structure much better than the coarser resolutions. Moreover, the velocity

5   deficit decays at a much slower rate in the flat case than the other (complex) cases. The velocity deficit for turbine located on second ridge is observed to be the smallest with the fastest recovery. This can be attributed to the highest ambient turbulence in the wake of the particular turbine among the considered cases. To support this argument, we plot normalized profiles of turbulent kinetic energy $TKE = \frac{\langle \sigma_u^2 + \sigma_v^2 + \sigma_w^2 \rangle}{2}$ (see Figure 11). As turbulent kinetic energy is responsible for transport of energy in the domain, it can be used to quantify the wake recovery. From the figure, it is clearly observed that the turbulent kinetic

10   energy for complex terrain cases is significantly higher than the flat terrain case. The turbine located on top of the second ridge in the steep case shows highest levels of TKE, thus causing fastest wake recovery.



## 5.4 Wake Self-Similarity

Figure 12 shows how the maximum velocity deficit develops as the wake moves in the downstream. An increasing-decreasing behavior can be observed, with the smooth case showing the highest velocity deficit. At a downstream distance of 5 rotor diameters, the three complex cases show very close numbers for maximum velocity deficit. The flat case (as expected) shows
slowest recovery in wake.

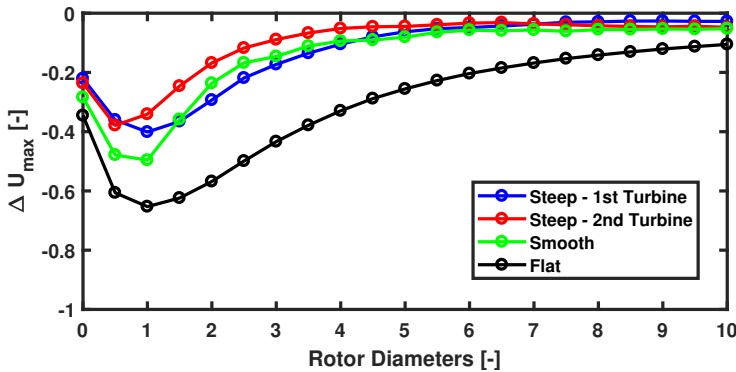

**Figure 12.** Maximum normalized mean velocity deficit as a function of downstream distance for different cases.

To further characterize the wake, we plot right and left wake half-widths for the lateral and vertical profiles in figure 13. The wake half-widths generally show an increasing trend with the downstream distance. For the lateral profiles the values are much closer near the turbine location and spread differently for different cases, whereas, for vertical profiles, the values have a wider spread due to varying levels of shear. The flat terrain is observed to have smallest wake half-widths, whereas, the 2nd
turbine in steep slope shows highest values. This can be attributed to the rate of wake recovery in the respective cases. Finally, the difference in the left and right wake widths for a specific location highlight the wake symmetry/asymmetry.

We now plot normalized velocity deficit profiles against normalized distance from wake center. The velocity deficit profiles are normalized with the maximum velocity deficit for the respective profiles, whereas, the centered distance is normalized with wake half-width on either side of the wake center.

Figure 14 (a) shows lateral self-similar velocity deficit profiles for the flat case. The profiles follow the Gaussian profile and last from a downstream distance of 1 to 9 rotor diameters. The same profiles for three complex cases shown in figure 14 (b) also collapse on the Gaussian profile, however, two important things should be noted. First the self-similarity holds for a much shorter distance (from 1 to 3 rotor diameters). This can be attributed to the faster wake recovery in complex cases, as well as, to the fact that the localized terrain changes disrupt the wake structure and thus deviations from self-similarity occur. Second
important feature is the wide spread in the tails of the profiles for three complex cases. Whereas, the profiles for a single case (denoted by same color) are much closer to each other, wider spread is seen while comparing profiles for two different cases.



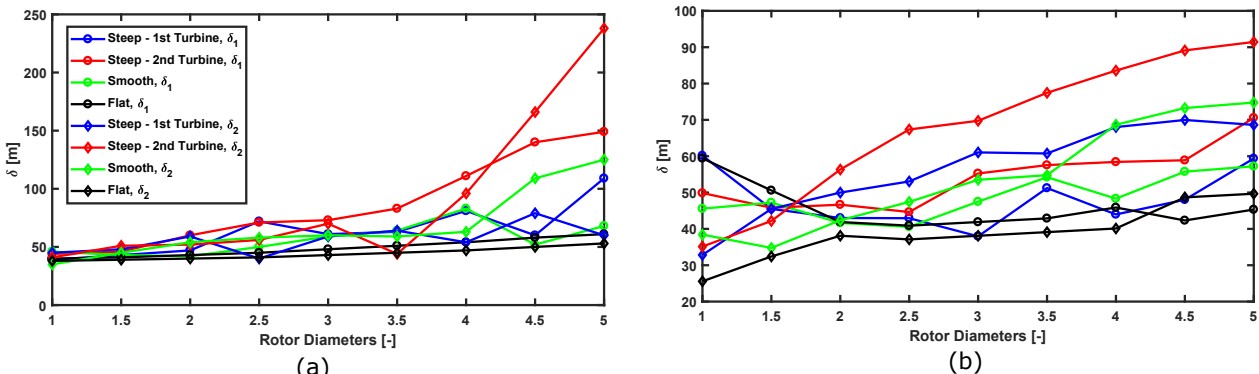

**Figure 13.** Right and left wake half-width for the (a) lateral and (b) vertical velocity deficit profiles.

This can be due to the difference in the shear toward the edges of the wakes in different cases, as well as, to the differences arising due to difference in terrains, flow separation characteristics and levels of turbulence.

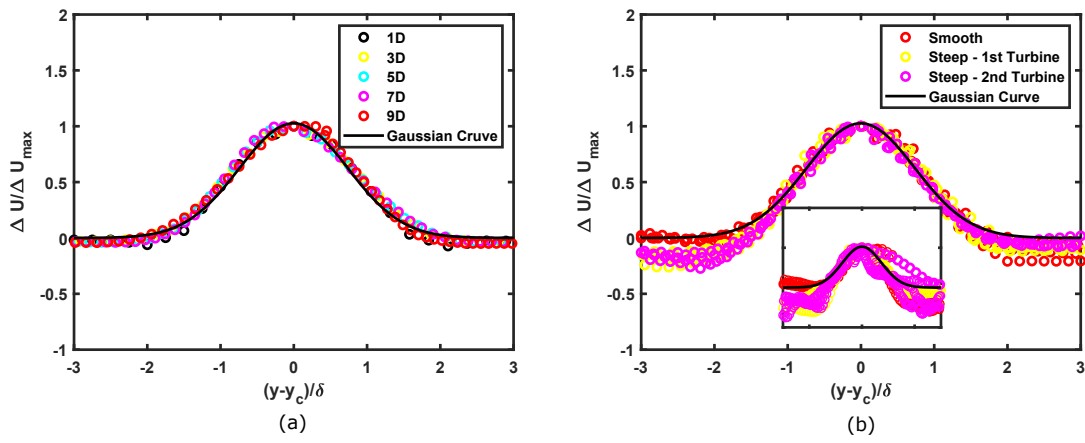

**Figure 14.** Self-similar lateral velocity deficit profiles (a) flat case (b) complex cases for a downstream distance of 1 D to 3 D with intervals of 0.5 D; inset: normalized velocity deficit profiles from 3.5 D to 5 D.

Figure 15 (a) shows vertical self-similar velocity deficit profiles for the flat case. Self-similarity is observed from a distance of 3 to 10 rotor diameters. In the complex cases (figure 15 (b)), self similarity in the vertical profiles is observed from 1 to 3 rotor diameters. These profiles show a much wider spread when compared to the lateral profiles. Moreover, they also exhibit an asymmetric behavior than the lateral profiles. This is due to the impact of surface and vertical wind shear. In addition, as the terrain is complex, varying levels of slope lead to varying distances between the projected rotor area and the surface, which eventually gives rise to high variation among profiles at different downstream locations.





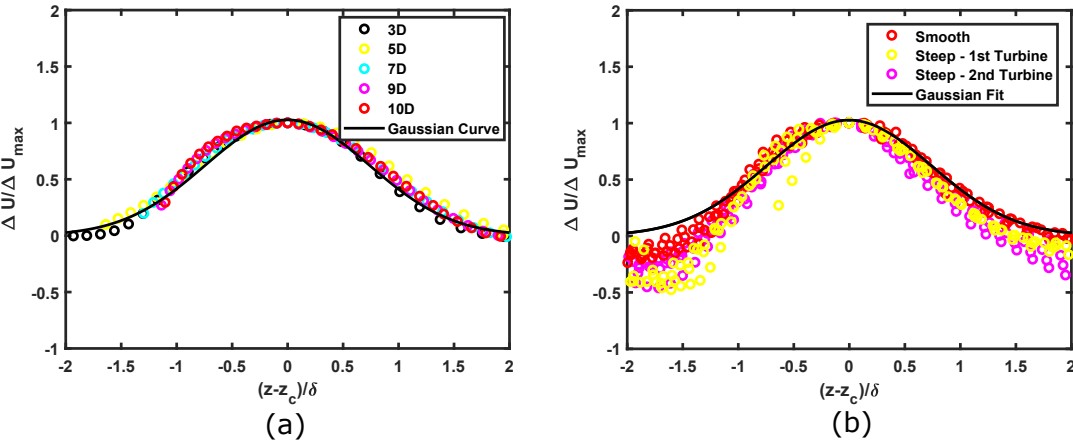

**Figure 15.** Self-similar vertical velocity deficit profiles (a) flat case (b) complex cases for a downstream distance of 1 D to 3 D with intervals of 0.5 D; inset: normalized velocity deficit profiles from 3.5 D to 5 D.

## 6 Conclusion

Terrain generated phenomenon can have significant impact on the wake characteristics of a turbine located in complex terrain. In this study, we attempted to verify whether the self-similarity, which is one the fundamental characteristics of wakes, still holds in complex terrain or not. In this context, we performed large-eddy simulation of wind turbine(s) located in a flat as well as complex terrain. By varying turbine location, as well as, terrain complexity, we simulated different flow scenarios in complex terrain. By plotting normalized velocity deficit profiles in the lateral, as well as, vertical direction, we looked for self-similarity in the simulated cases. We observed that wakes in complex terrain preserve self-similarity in both directions. The region of this preservation is, however, over much shorter distances downstream than the flat terrain counter-part. This is attributed to the deformation of wakes in the far wake region by varying heights of terrain, as well as, to the faster wake recovery in the complex terrain. Finally, vertical profiles show a wider spread and higher asymmetry than the lateral ones.

*Acknowledgements.* The authors would like to thank Niels N. Sørensen for help in preparing the surface meshes and Peter P. Sullivan, National Center for Atmospheric Research, Boulder, CO, US, for sharing the LES code.



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
