# Peer review of "On the self-similarity of wind turbine wakes in complex terrain using large-eddy simulation"

_Wind Energy Science, 2019_

## Referee Comment (RC1) · Dries Allaerts (Referee) · 13 Jun 2019

This paper performs large-eddy simulations over the complex terrain at Perdigao and investigates the impact of terrain on the wakes of wind turbines. Overall, the authors did a good job in selecting relevant cases to analyse the impact of increasingly complex terrain on turbine wakes, and appropriate normalizations are performed to be able to compare self-similar behavior under very different flow conditions. I believe the results are of interest to the wind energy community and important for future development of wake models in complex terrain. I do have some scientific questions and minor comments as listed below.

Scientific questions/issues

[Figure]

1. Page 1 line 13-15. You seem to suggest that wind turbine wakes in complex terrain are getting more attention because wind farm development is shifting toward complex terrain as a result of the depletion of flat terrain sites. Can you provide a reference for this statement?

2. Section 2. I appreciate that the main aspects of the LES framework are summarized in the paper and that the reader is referred to Sullivan et al. (2014) for more details. However, upon reading Sullivan's paper I noticed some differences in the formulation, for example in Eq. 3-5 compared to Sullivan's Eq. 2. I would try to use the exact same notation to avoid confusing the reader. Moreover, Sullivan says that $U_i$ is normal to surfaces of constant $\xi_i$, while you say this velocity is normal to $x_i$ surfaces (page 4 line 1). I assume this is a typo?

3. Section 2.1. First, actuator disk models have been used in many LES studies, so some references to pioneering work should be provided. Second, using the disk velocity instead of the free stream velocity when the latter is not well defined was first introduced by Calaf et al. (2010) so I would refer to that paper instead of Hansen (2015). Third, you seem to suggest that the velocity used to compute the thrust force is ensemble averaged, but I don't see how that could be implemented in a time-dependent LES framework. Rather, I believe the averaging denotes averaging over the rotor disk at every time step, like in Calaf et al. (2010). Please clarify. Finally, on line 24 of page 4, you seem to suggest that the induction and thrust coefficient are chosen to be 1/4 and 3/4, but in fact these two quantities are related by one-dimensional momentum theory, so an induction of 1/4 implies a thrust coefficient of 3/4.

4. Page 5 line 13-14. A turbine is mentioned in the case setup but it is not mentioned where the turbine is located. Likewise, table 1 shows some simulations have two turbines, but the placement of the turbines is not documented. Although the location can be deduced later in the paper, it would be best to give this information

in the setup section.

5. Section 3.2. I don't understand how averaging 40 (or any other number of) thirty-minute-averaged LES fields is different from just averaging over the total 20 hours. It feels to me that section 3.2 can simply be replaced by stating that the LES results are averaged over a number of hours to reach statistical convergence. Further, I wonder how the odd number of 30-minute windows in table 1 have been chosen per simulation. Did you estimate how long you needed to average to reduce the statistical convergence error below a certain limit?

6. Section 5.1 In a periodic domain there is not really an inflow velocity or a "first grid point" (page 9 line 11). It makes more sense to specify what distance upstream of the first ridge you take as the inflow velocity.

7. Page 10 line 4. What do you mean with the relative positioning of the rotor in the two cases? How are they different? Wouldn't it make more sense to have the rotors be at the exact same horizontal and vertical location independent of the grid resolution?

Other minor editorial changes

- The term stationarity is throughout the paper when discussing ensemble averaging, but I think the authors mean statistical convergence.

- Page 4 L24: A reference is missing and replaced with (?)

- Page 7 L3-4: Is the domain height $H$ the same as $Z_L$ in Eq. 9?

- Figures 5,6,10,11: is $h$ the same as $h_L$ in Eq. 9?

- Page 10 L26: "behind the three turbines" should I guess be "behind the two turbines".

- Figure 8: use z instead of h to indicate height to be consistent with other figures. Also, consider reordering figure 8b to have the smooth case on top so as to match the order of 8a

- Page 15 line 6: a "more" asymmetric behavior than the lateral profiles.

- Caption of figure 15: There is no inset in this figure.

---

## Referee Comment (RC2) · Anonymous Referee #2 · 9 Jul 2019

**Review of "On the self-similarity of wind turbine wakes in complex terrain using large-eddy simulation", by A. S. Dar, J. Berg, N. Troldborg, E. G. Patton**

This paper studies self-similarity of turbine wakes sited on complex terrain using large eddy simulation (LES). Five simulations with varying surface features and number of turbines are performed. Basic features of wake-terrain interaction, that have been observed previously, are reaffirmed here. These include horizontal and vertical deflection of the wake centerline, and sensitivity of the turbulent kinetic energy and rate of wake recovery to terrain. The primary novelty of this paper is assessing whether and under what conditions the mean velocity deficit shows self-similarity.

This is an important topic and is of use since self-similarity is an underlying assumption in several analytical models. The paper is well-written and clear to follow. There are a few issues related to grid independence, where self-similarity is studied, and self-similarity of quantities other than the mean velocity deficit, that should be clarified to further improve this manuscript. Please see below for points that should be further clarified.

**Major Issues:**

1. The authors establish grid independence of their results by comparing two cases with different grid sizes for the same steep topography (probably cases 3 and 4 in Table 1). As the grid is refined, more features of the topography are resolved. The differences in terrain height between the coarsely sampled and the finely sampled cases seem to be about 30 m (from Fig. 6, panel 1D), which is quite large compared to the 80 m turbine diameter. As a result, the differences between cases 3 and 4 mentioned in the paper (e.g. the 15% difference shown in Fig. 7) can be ascribed partly to grid size and partly to differing terrain. Given this, a grid independence study should be performed with terrain remaining 'frozen' across grid resolutions. This will enable the authors to comment exclusively on the influence of the grid without contamination by the influence of terrain.

2. It isn't clear from the text in which horizontal and vertical planes the self-similarity is being evaluated. Specifically, what are the elevation (z) values for the profiles corresponding to 1D, 2D and 3D downstream of the turbine in Fig. 14? Also, what spanwise locations (y) are being referred to in Fig. 15? These questions arise because the wake centerline deflects in both vertical and spanwise directions, and I suspect that the observations regarding collapse of profiles at different x locations onto a single curve might be sensitive to the planes selected for this analysis.

3. It would be interesting to check if the behavior of a turbine wake that deflects vertically (and laterally) is comparable to other free-shear flows that deflect in this manner. An example is a horizontal buoyant jet studied in Xu & Chen (2012), but there might be other studies as well, such as a jet in cross-flow. The asymmetry displayed in Fig. 15 (b) seems

similar to Fig. 15 in Xu & Chen (2012). Are there any systematic trends in the deviation from Gaussian profile in a turbine wake?

4. It would be interesting to check for self-similarity of other quantities such as RMS of fluctuations and other components of the Reynolds stress tensor (e.g. $\overline{u'w'}$). This is important because of some recent analytical models that rely on the self-similarity of the added TKE (e.g. Ishihara & Qian, 2018).

**Minor Issues:**

1. Section 2.1, Line 25: References missing here.
2. Fig. 1 indicates that the turbines were not at the center of the domain in the spanwise (y) direction. Why were the spanwise extents chosen in this manner?
3. Was the topography naturally periodic over the chosen extents in the x and y directions, or were some artificial adjustments to the topography introduced to ensure periodicity in x and y?
4. Section 3, first paragraph: 'H' is used without being defined.
5. Page 6, Line 8: "…chosen for this terrain to avoid Gibbs phenomenon." Can the authors explain this sentence in a little more detail? Is it that the terrain without any smoothing at all leads to Gibbs oscillations in the simulation?
6. Page 6, Line 14: Is there a reference to a systematic study where the 1:4 aspect ratio being suitable is demonstrated?
7. Section 3.1, Line 5: Can the authors add the resulting friction velocity values to Table 1? This way, the differences in the terrain-induced drag forces will become apparent.
8. Section 3.1, Line 10: Does the wall model need to be tweaked for correctly handling the recirculation regions on the lee-side of complex terrain? The authors mention in Section 5.3 that such recirculation regions exist in the current simulations.
9. Section 3.2: This is slightly confusing. Are the 30-minute averages from the same simulation or from different simulations? If, say, five 30-minute ensembles are used, how is the resulting average velocity different from a 150-minute average? How is each individual 30-minute average computed: i.e., averaged using fields at each time step, or every few time steps?
10. Fig. 5: By what amount do the $u_h$ values vary? I assume they would be proportional to the respective $u_*$ values, so it would help to have these tabulated along with the friction velocities as mentioned in point 12.
11. Please mention what grid sizes are being used for all cases in Fig. 5.
12. Section 5.1, last line: I can understand the horizontal heterogeneity introduced by the complex terrain being responsible for deviation from log-law, but I do not understand how the boundary conditions affect this. Could the authors clarify why periodicity could be responsible for deviations from log-law?
13. Page 11, last paragraph: Do the authors suggest here that the spanwise deflection of the wakes could be a numerical artifact? It should be straight-forward to determine if the value of the spanwise gradient of the terrain is significant at these locations.

14. Section 5.3.1, lines 1-3: I assume 'lateral profiles' are referring to Fig. 9, and that the steep case uses finer resolution and smooth case uses coarse resolution. What additional features are captured by finer resolution (steep case) in this figure that are not seen in the coarser resolution (smooth case) simulation?

15. Fig. 12: Please clarify why the velocity deficits are so different at x/D = 0. One can understand their evolution being different, but at the disk, these quantities should be very close to each other. Are these differences related to the reference velocity?

16. Fig. 14: The authors should make the inset as a separate subpanel (Fig. 14c) so as to show clearly that self-similarity does not seem to hold beyond 3D.

17. Fig. 15: The caption mentions an inset, but it is missing from the figure.

18. Minor stylistic issues:
   a. Please ensure that the references are in an appropriate order.
   b. Section 2, Line 1: Remove brackets from "formulated in (Sullivan et al. 2014)"
   c. If possible, the authors should use the same color scheme for the different cases in all figures.
   d. Fig. 8: It would be easier to read this if the order between (a) and (b) were to be maintained, i.e. 'Smooth' on the top and 'Steep' below.

**References:**

T. Ishihara, G. Qian (2018): "A new Gaussian-based analytical wake model for wind turbines considering ambient turbulence intensities and thrust coefficient effects", *Journal of Wind Engineering and Industrial Aerodynamics*, 177, 275 – 292.

D. Xu, J. Chen (2012): "Experimental study of stratified jet by simultaneous measurement of velocity and density fields", *Experiments in Fluids*, 53, 145 – 162.

---

## Author Comment (AC1) · 14 Aug 2019

**Author reply to reviewers on "On the self-similarity of wind turbine wakes in complex terrain using large-eddy simulation" by Arslan Salim Dar et al.**

**June 2019**

We thank the reviewer, Dries Allaerts, and the anonymous reviewer for the nice words and constructive criticism which have helped in making our paper better and more readable. We have addressed their questions and issues/concerns below in red.

**Review report from Dries Allaerts**

**1**

Page 1 line 13-15. You seem to suggest that wind turbine wakes in complex ter- rain are getting more attention because wind farm development is shifting toward complex terrain as a result of the depletion of flat terrain sites. Can you provide a reference for this statement? for example : this ans that reference is added

*Although, we cannot find a study which provides statistics of wind energy development in flat or complex terrains, the argument that complex terrains are getting attention partly due to immense development of wind farms in flat terrains has been used in various studies before, e.g. (Alfredsson and Segalini, 2017) state: "With complex terrain, we mean features of the Earth's surface that influence the wind in the atmospheric boundary layer (ABL) through terrain topology and/or roughness effects (figure 1). Historically, these have been avoided because of the harsher wind conditions (both lower wind speeds and higher turbulence intensity are expected), but are now becoming more appealing from the wind industry's point of view due to the lack of better alternatives." Similarly, (Feng et al., 2017) state: "For onshore wind farm constructions, since a lot of good sites in flat terrain have already been occupied, more attentions are paid to complex terrain sites, especially for countries featured by a large percentage of mountainous terrain, such as China."*

**2**

Section 2. I appreciate that the main aspects of the LES framework are summa- rized in the paper and that the reader is referred to Sullivan et al. (2014) for more details. However, upon reading Sullivan's paper I noticed some differences in the formulation, for example in Eq. 3-5 compared to Sullivan's Eq. 2. I would try to use the exact same notation to avoid confusing the reader. Moreover, Sullivan says that Ui is normal to surfaces of constant i, while you say this velocity is normal to xi surfaces (page 4 line 1). I assume this is a typo? Yes, that is a typo. We have corrected the text and changed notation to fit Sullivan 2014.

**3**

Section 2.1. First, actuator disk models have been used in many LES studies, so some references to pioneering work should be provided. Second, using the disk velocity instead of the free stream

velocity when the latter is not well defined was first introduced by Calaf et al. (2010) so I would refer to that paper instead of Hansen (2015). Third, you seem to suggest that the velocity used to compute the thrust force is ensemble averaged, but I don't see how that could be implemented in a time-dependent LES framework. Rather, I believe the averaging denotes averaging over the rotor disk at every time step, like in Calaf et al. (2010). Please clarify. Finally, on line 24 of page 4, you seem to suggest that the induction and thrust coefficient are chosen to be 1/4 and 3/4, but in fact these two quantities are related by one-dimensional momentum theory, so an induction of 1/4 implies a thrust coefficient of 3/4. Yes, you are right in all points and we have added Calaf et al.

**4**

Page 5 line 13-14. A turbine is mentioned in the case setup but it is not mentioned where the turbine is located. Likewise, table 1 shows some simulations have two turbines, but the placement of the turbines is not documented. Although the location can be deduced later in the paper, it would be best to give this information in the setup section.

*Added the information on turbine location in the respective places.*

**5**

Section 3.2. I don't understand how averaging 40 (or any other number of) thirty-minute-averaged LES fields is different from just averaging over the total 20 hours. It feels to me that section 3.2 can simply be replaced by stating that the LES results are averaged over a number of hours to reach statistical convergence. Further, I wonder how the odd number of 30-minute windows in table 1 have been chosen per simulation. Did you estimate how long you needed to average to reduce the statistical convergence error below a certain limit? We chose to do it this way in order to stick to the "convention" of using 30 min averaging time when estimating fluxes in micro meteorological applications. This also allows us to create error bars - since each 30 min is not identical. But yes a long term average is of course equal to the mean indicated in equation 16. the number of ensemble used is a little random. A server breakdown forced us to stop at the given numbers.

**6**

Section 5.1 In a periodic domain there is not really an inflow velocity or a "first grid point" (page 9 line 11). It makes more sense to specify what distance upstream of the first ridge you take as the inflow velocity.

*Changed the description as per reviewer's suggestion.*

**7**

Page 10 line 4. What do you mean with the relative positioning of the rotor in the two cases? How are they different? Wouldn't it make more sense to have the rotors be at the exact same horizontal and vertical location independent of the grid resolution?

*The turbines are located at the exact location in both cases, this is evident from comparing the velocity profiles in the two terrains at the turbine location (Figure 6, panel 3 (At Disk)). Due to change in slope in the two cases, the distance between a fixed point in the vertical direction and the terrain surface changes as we move down (or up) stream of the ridge top in the two terrains. We agree that mentioning "relative positioning of the rotor" might be confusing and therefore replace it with better description in page 10 line 4 and line 10.*

**0.1 Other minor editorial changes**

- The term stationarity is throughout the paper when discussing ensemble averag- ing, but I think the authors mean statistical convergence. for example : done

- Page 4 L24: A reference is missing and replaced with (?) *Added*

- Page 7 L3-4: Is the domain height H the same as ZL in Eq. 9? Yes, and corrected

- Figures 5,6,10,11: is h the same as hL in Eq. 9? yes, and corrected

- Page 10 L26: "behind the three turbines" should I guess be "behind the two turbines". *Corrected*

- Figure 8: use z instead of h to indicate height to be consistent with other figures. Also, consider reordering figure 8b to have the smooth case on top so as to match the order of 8a *Done*

- Page 15 line 6: a "more" asymmetric behavior than the lateral profiles. *Added*

- Caption of figure 15: There is no inset in this figure. *Added*

**Review report from anonymous reviewer**

**1**

The authors establish grid independence of their results by comparing two cases with different grid sizes for the same steep topography (probably cases 3 and 4 in Table 1). As the grid is refined, more features of the topography are resolved. The differences in terrain height between the coarsely sampled and the finely sampled cases seem to be about 30 m (from Fig. 6, panel 1D), which is quite large compared to the 80 m turbine diameter. As a result, the differences between cases 3 and 4 mentioned in the paper (e.g. the 15% difference shown in Fig. 7) can be ascribed partly to grid size and partly to differing terrain. Given this, a grid independence study should be performed with terrain remaining 'frozen' across grid resolutions. This will enable the authors to comment exclusively on the influence of the grid without contamination by the influence of terrain. you are Right, but we did not have access to unlimited cpu resources so we had to make a choice. We have added a comment about this fact i nthe paper.

**2**

It isn't clear from the text in which horizontal and vertical planes the self-similarity is being evaluated. Specifically, what are the elevation (z) values for the profiles corresponding to 1D, 2D and 3D downstream of the turbine in Fig. 14? Also, what spanwise locations (y) are being referred to in Fig. 15? These questions arise because the wake centerline deflects in both vertical and spanwise directions, and I suspect that the observations regarding collapse of profiles at different x locations onto a single curve might be sensitive to the planes selected for this analysis. For the lateral profiles, we initially checked for self-similarity taking a horizontal plane (xy) passing through the respective turbine hub heights. This is a simple approach and gives an idea of the self-similarity in one single plane. We have accounted for the wake deflection in the vertical direction. The resulting finding shows an improvement over the single plane approach, however, beyond 3.5 D the tails are still observed to significantly depart from Gaussian profile, complementing the earlier finding. For the sideways deflection, the deflection we observed in the maximum velocity deficit in spanwise direction was within one or two grid points and therefore we just use the vertical plane (xz) passing through the turbine hub position.

**3**

It would be interesting to check if the behavior of a turbine wake that deflects vertically (and laterally) is comparable to other free-shear flows that deflect in this manner. An example is a horizontal buoyant jet studied in Xu  Chen (2012), but there might be other studies as well, such as a jet in cross-flow. The asymmetry displayed in Fig. 15 (b) seems similar to Fig. 15 in

Xu & Chen (2012). Are there any systematic trends in the deviation from Gaussian profile in a turbine wake? A systematic trend is indeed observed in the vertical self-similar profiles, where, with increasing downstream distance, the deviation from Gaussian profile increases on both tails of the profiles. This has now been indicated in the figure 16 (b). We refrain from making any direct comparison with other free-shear flows (for now), as the physical phenomenon responsible for these deviations can be different for each of such cases and thus require a careful and systematic comparison.

**4**

It would be interesting to check for self-similarity of other quantities such as RMS of fluctuations and other components of the Reynolds stress tensor (e.g. $\overline{u'w'}$ ). This is important because of some recent analytical models that rely on the self-similarity of the added TKE (e.g. Ishihara & Qian, 2018). looking into flux budgets and the Reynolds stress tensor is indeed interesting and could provide valuable insight. We, however, consider the outside the scope of the present paper.

**Minor issues**

1. Section 2.1, Line 25: References missing here. added

2. Fig. 1 indicates that the turbines were not at the center of the domain in the spanwise (y) direction. Why were the spanwise extents chosen in this manner? The turbine location is picked to match the location of a turbine placed in the terrain in reality

3. Was the topography naturally periodic over the chosen extents in the x and y directions, or were some artificial adjustments to the topography introduced to ensure periodicity in x and y? Minimum changes were made in the streamwise direction - a few more in the spanwise direction. We have added a note about this in the paper.

4. Section 3, first paragraph: 'H' is used without being defined. $H$ is now defined in the section II

5. Page 6, Line 8: "...chosen for this terrain to avoid Gibbs phenomenon." Can the authors explain this sentence in a little more detail? Is it that the terrain without any smoothing at all leads to Gibbs oscillations in the simulation? yes and the code becomes numerical unstable at these cell sizes, just a tiny smothering avoids this.

6. Page 6, Line 14: Is there a reference to a systematic study where the 1:4 aspect ratio being suitable is demonstrated? We dont know of any - and hence have added a "rule-of-thumb" statement.

7. Section 3.1, Line 5: Can the authors add the resulting friction velocity values to Table 1? This way, the differences in the terrain-induced drag forces will become apparent. relevant pressure data is not available to compute an accurate measure of change in friction velocity.

8. Section 3.1, Line 10: Does the wall model need to be tweaked for correctly handling the recirculation regions on the lee-side of complex terrain? The authors mention in Section 5.3 that such recirculation regions exist in the current simulations. No, tweaking to the wall model has been applied: a log-law in the first cell value acroos the domain and locally evaluated is used.

9. Section 3.2: This is slightly confusing. Are the 30-minute averages from the same simulation or from different simulations? If, say, five 30-minute ensembles are used, how is the resulting average velocity different from a 150-minute average? How is each individual 30-minute average computed: i.e., averaged using fields at each time step, or every few time steps? We chose to do it this way in order to stick to the "convention" of using 30 min averaging time when estimating fluxes / turbulence in micro meteorological applications. This also allows

us to create error bars - since each 30 min is not identical. But yes a long term average is of course equal to the mean indicated in equation 16.

10. Fig. 5: By what amount do the uh values vary? I assume they would be proportional to the respective u* values, so it would help to have these tabulated along with the friction velocities as mentioned in point 12. added information on the reference velocities in section 4.

11. Please mention what grid sizes are being used for all cases in Fig. 5. done

12. Section 5.1, last line: I can understand the horizontal heterogeneity introduced by the complex terrain being responsible for deviation from log-law, but I do not understand how the boundary conditions affect this. Could the authors clarify why periodicity could be responsible for deviations from log-law? terrain generated flow phenomenon could re-enter from the other side, e.g. the far wake from the ridges themselves.

13. Page 11, last paragraph: Do the authors suggest here that the spanwise deflection of the wakes could be a numerical artifact? It should be straight-forward to determine if the value of the spanwise gradient of the terrain is significant at these locations. the sideways deflection in wake center was within one or two grid points, so we do not make any conclusive statement on it.

14. Section 5.3.1, lines 1-3: I assume 'lateral profiles' are referring to Fig. 9, and that the steep case uses finer resolution and smooth case uses coarse resolution. What additional features are captured by finer resolution (steep case) in this figure that are not seen in the coarser resolution (smooth case) simulation? color We have only focused on features related to the wake characteristics within the statistics presented in the paper. Looking at profiles of other statistics we believe other features to show up. at this stage we, however, feel that that would be work beyond the present paper.

15. Fig. 12: Please clarify why the velocity deficits are so different at x/D = 0. One can understand their evolution being different, but at the disk, these quantities should be very close to each other. Are these differences related to the reference velocity? yes, the reference velocities can be different in different cases

16. Fig. 14: The authors should make the inset as a separate subpanel (Fig. 14c) so as to show clearly that self-similarity does not seem to hold beyond 3D. we have added the inset into the main figure for vertical profiles and indicated the direction of increasing downstream distance which helps visualize when the departing of tails from Gaussian become significant. In addition, added a new figure to complement the discussion on lateral profiles.

17. Fig. 15: The caption mentions an inset, but it is missing from the figure. modified

18. Minor stylistic issues:

    a . Please ensure that the references are in an appropriate order. this is now in alphabetic order.

    b . Section 2, Line 1: Remove brackets from "formulated in (Sullivan et al. 2014)" done

    c . If possible, the authors should use the same color scheme for the different cases in all figures. done

    d Fig. 8: It would be easier to read this if the order between (a) and (b) were to be maintained, i.e. 'Smooth' on the top and 'Steep' below. corrected

**References**

[1] P. H. Alfredsson, A. Segalini. Introduction Wind farms in complex terrains: an introduction. Philos Trans A Math Phys Eng Sci. 2017;375(2091):20160096. doi:10.1098/rsta.2016.0096

[2] J. Feng, W. Z. Shen, K. S. Hansen, A. Vignaroli, ... Liu, W. Wind farm design in complex terrain: the FarmOpt methodology. Paper presented at China Wind Power 2017, Beijing, China.

---

## Referee Report (RR1)

**Review of "On the self-similarity of wind turbine wakes in complex terrain using large-eddy simulation", by A. S. Dar, J. Berg, N. Troldborg, E. G. Patton**

The authors have responded to some of the comments raised previously but not all. As noted below, the responses in the response letter do not match with the revised manuscript. It may be that the authors uploaded the wrong file. The outstanding issues are quite minor, but I think they would improve readability of this paper for future readers.

1. **On response to "Major Issue 2":** The authors have responded to my question raised in 'Major Issue 2', but (unless I have missed them) haven't made any changes to the manuscript. So, from the manuscript, it still isn't clear in which horizontal and vertical planes the self-similarity is being evaluated. Please include this information in the manuscript so as to help future readers.

2. **On response to "Major Issue 3":** I do not see a Figure 16(b) in the revised manuscript. Did the authors mean Figure 15(b) in their response?

3. **On response to "Minor Issue 1":** The authors claim to have added a reference here, but in fact I don't see it in the revised paper.

4. **On response to "Minor Issue 10":** The authors mention that the value of $u_h$ has been written in Section 4, but I again cannot find it in Section 4 of their revised manuscript.

5. **On response to "Minor Issue 11":** The authors have claimed to have mentioned grid sizes of the cases shown in Fig. 5, but I do not see it in the caption or in the text accompanying Fig. 5. I understand that this information is mentioned in Table 1. But there are three cases labled 'Steep' in Table 1, and it is unclear which of these is depicted in Fig. 5.

6. **On response to "Minor Issue 15":** If the differences between velocity deficits are caused by the normalization, the values used for normalization should be mentioned. Going back to point 4 above, I could not find the reference velocities $u_h$ anywhere in the manuscript or in Section 4 as the authors claimed in their response.

7. **On response to "Minor Issue 16":** The authors refer to an additional figure in their response, but I do not see an additional figure in their manuscript.

---

## Author Response (AR2)

**Review of "On the self-similarity of wind turbine wakes in complex terrain using large eddy simulation", by A. S. Dar, J. Berg, N. Troldborg, E. G. Patton**

The authors have responded to some of the comments raised previously but not all. As noted below, the responses in the response letter do not match with the revised manuscript. It may be that the authors uploaded the wrong file. The outstanding issues are quite minor, but I think they would improve readability of this paper for future readers.

It does appear to me that either we uploaded the wrong version of manuscript or the reviewer read the older version of manuscript, as ALL the below-mentioned points were already addressed in our revised manuscript.

1. On response to "Major Issue 2": The authors have responded to my question raised in 'Major Issue 2', but (unless I have missed them) haven't made any changes to the manuscript. So, from the manuscript, it still isn't clear in which horizontal and vertical planes the self-similarity is being evaluated. Please include this information in the manuscript so as to help future readers.

Refer to lines 15-22, page 15

2. On response to "Major Issue 3": I do not see a Figure 16(b) in the revised manuscript. Did the authors mean Figure 15(b) in their response?

Figure 16 is added to the manuscript, top page 17.

3. On response to "Minor Issue 1": The authors claim to have added a reference here, but in fact I don't see it in the revised paper.

Reference is added in the mentioned line.

4. On response to "Minor Issue 10": The authors mention that the value of uh has been written in Section 4, but I again cannot find it in Section 4 of their revised manuscript.

Added on Page 9, top.

5. On response to "Minor Issue 11": The authors have claimed to have mentioned grid sizes of the cases shown in Fig. 5, but I do not see it in the caption or in the text accompanying Fig. 5. I understand that this information is mentioned in Table 1. But there are three cases labeled 'Steep' in Table 1, and it is unclear which of these is depicted in Fig. 5.

Grid size is mentioned in caption of figure 5.

6. On response to "Minor Issue 15": If the differences between velocity deficits are caused by the normalization, the values used for normalization should be mentioned. Going back to point 4 above, I could not find the reference velocities uh anywhere in the manuscript or in Section 4 as the authors claimed in their response.

Added on Page 9, top.

7. On response to "Minor Issue 16": The authors refer to an additional figure in their response, but I do not see an additional figure in their manuscript.

Added on Page 17

---

## Author Response (AR3)

We thank the editor for these technical comments.

"det" in eq 6 should be roman.
Corrected
Shouldn't "phenomenon" in first line of conclusion be plural "phenomena"?
Corrected
Don't use "2nd" but "second" (p11)
We believe the page numbering here is wrong – but corrected in the text on p13. We still use the shorter version 2$^{nd}$ in captions and figure texts
Fig 14 a caption last line "cruve" -> "curve" (same with fi 15 b)
Corrected
I can't see the definition of "self similar" very early in the paper. I think it should be there very early, just one line like "same shape as a function of downstream distance when scaled with maximum deficit and width..." (if this is correct).
Added in the introduction, Thanks for the advice.
Please use the Copernicus WES latex style file with the additional info at the end of the paper about data/code availability etc.
Added
References:
2 "Port" ->"Porte" (same with ref 6)
Corrected
21 ~ in Pena should be over n, not e.
Corrected
"Vasiljevic" spelled wrong in ref 23 and 33.
Corrected
Is Archer C of C L in refs 35 and 36?
This is how it appears in the respective papers. In one it is C L Archer, in other it is C Archer

On behalf of the authors
Jacob Berg, DTU Wind Energy